# SAViT: Structure-Aware Vision Transformer Pruning via Collaborative Optimization

**Chuanyang Zheng**[1]**, Zheyang Li**[1,2]**, Kai Zhang**[1]**, Zhi Yang**[1]**,**
**Wenming Tan**[1]**, Jun Xiao**[2]**, Ye Ren**[1]**, Shiliang Pu**[1]
[1] Hikvision Research Institute, Hangzhou, China
[2] Zhejiang University, Hangzhou, China
`{zhengchuanyang,lizheyang,zhangkai23,yangzhi13}@hikvision.com`
`tanwenming@hikvision.com, junx@cs.zju.edu.cn`
`{renye,pushiliang.hri}@hikvision.com`

## Abstract

Vision Transformers (ViTs) yield impressive performance across various vision tasks. However, heavy computation and memory footprint make them inaccessible for edge devices. Previous works apply importance criteria determined independently by each individual component to prune ViTs. Considering that heterogeneous components in ViTs play distinct roles, these approaches lead to suboptimal performance. In this paper, we introduce joint importance, which integrates essential structural-aware interactions between components for the first time, to perform collaborative pruning. Based on the theoretical analysis, we construct a Taylor-based approximation to evaluate the joint importance. This guides pruning toward a more balanced reduction across all components. To further reduce the algorithm complexity, we incorporate the interactions into the optimization function under some mild assumptions. Moreover, the proposed method can be seamlessly applied to various tasks including object detection. Extensive experiments demonstrate the effectiveness of our method. Notably, the proposed approach outperforms the existing state-of-the-art approaches on ImageNet, increasing accuracy by 0.7% over the DeiT-Base baseline while saving 50% FLOPs. On COCO, we are the first to show that 70% FLOPs of Faster R-CNN with ViT backbone can be removed with only 0.3% mAP drop. The code is available at `https://github.com/hikvision-research/SAViT`.

## 1 Introduction

Convolutional neural networks (CNNs) have dominated nearly every aspect of computer vision for a long time [1]. Recently, the emerging ViTs [2, 3, 4, 5, 6, 7] have shown competitive performance on image classification, object detection, and other vision tasks. Despite the success, ViTs contain complicated submodules and require more intensive computational cost [8], limiting their deployment on resource-restricted devices. This naturally calls for compression and acceleration in ViTs as those done in CNNs.

One common direction for speeding up deep neural networks is to remove a group of less important connections that have a negligible impact on the network's performance. As pruning is proven very powerful to accelerate CNNs [9, 10, 11, 12], the exploration of pruning ViTs has just emerged. Different from CNNs whose parameters mainly rely on the homogeneous component, i.e., convolutional filter, transformers contain heterogeneous components such as multi-head self-attention (MSA), hidden neurons, and embedding neurons, as shown in Figure 1. These components play different functional roles in capturing contextual information and each one is associated with a distinct

36th Conference on Neural Information Processing Systems (NeurIPS 2022).

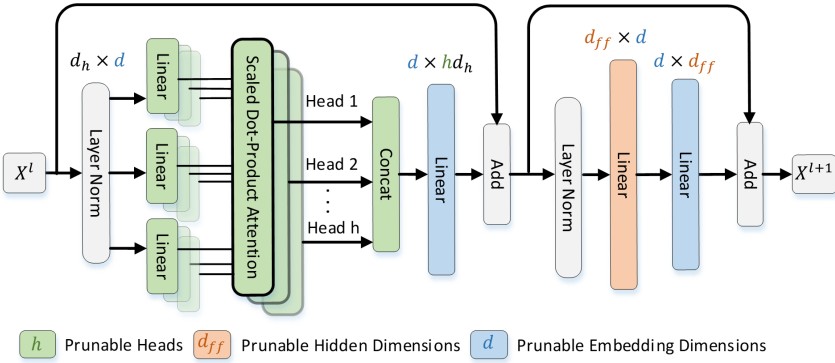

Figure 1: The illustration of prunable components in a ViT block. $d_h$ denotes the feature dimension in each head. ViTs consist of multiple repeated blocks.

structure. This makes pruning ViTs more challenging. Many previous approaches focus on trimming either heads [13] or patches [14, 15], while neglecting the time-consuming embedding component that accounts for the majority of redundancy as shown in our analysis. Most recent works [16, 17] aim at pruning multiple components in ViTs and gain better acceleration than pruning a single component. However, this line of methods ignores complicated interactions between components, leading to suboptimal compressed models. As pointed out by some studies [18] that worked on the interpretability of DNNs, interactions make meaningful contributions to the inference. Thus, interactions should be taken into account to minimize the impact of pruned parameters in ViTs.

In this work, we explore efficient ViTs through pruning all components comprehensively, which offers better flexibility to search for the optimal subnetwork in parameter design space and potentially reaches maximum acceleration for ViTs. Since all components collaborate with each other to achieve superior modeling capability instead of playing individually, we propose to quantitatively analyze the joint importance of pruned components, which contains the individual importance of each component as well as the interactions between components. Although preceding approaches including OBD [19] and OBS [20] have studied the interactions in pruning CNNs, the contribution of interactions is limited in CNNs [10], which consist of homogeneous components. Nevertheless, this is not the case for ViT. In this paper, we investigate the interactions between components and demystify that interactions are crucial in heterogeneous ViTs. Specifically, we propose to exploit the Hessian matrix to capture the interactions between components. To reduce the high computational cost of the Hessian matrix and facilitate the proposed algorithm for more applications, we encode the interactions represented by the Hessian matrix into pruning ratios of different components to efficiently approximate the optimization target. Finally, the Evolutionary Algorithm [21] is leveraged to solve the optimization problem. The proposed collaborative pruning algorithm is fast and consumes less than 2 epochs on the whole training dataset. In addition, the pruning framework can be seamlessly extended to accelerate more complicated networks such as detection networks. For example, by regarding the neck and head as new components, we can prune all components in the detection network together. We summarize our main contributions as follows:

- To the best of our knowledge, this work is the first to explicitly incorporate the interactions between different components into pruning ViTs.

- Based on the theoretical analysis, we transform the joint importance into an approximated optimization target to prune efficiently.

- We conduct a systematical analysis of ViTs and propose a comprehensive framework for pruning all components in ViTs by solving an optimization problem. The framework can also generalize well to more complicated networks such as detection networks, giving it the potential for widespread applications.

- The experimental results on various models and tasks demonstrate that the proposed approach brings great acceleration while preserving excellent performance. On ImageNet, our method can achieve a surprising 0.7% accuracy increase over DeiT-Base while reducing 50% FLOPs. On COCO, 70% FLOPs of the Faster R-CNN with Swin-T backbone can be removed with only 0.3% mAP drop.

## 2  Related Work

### 2.1  Vision Transformers

Transformer [22] is originally designed for natural language processing (NLP) tasks. It relies on self-attention to capture long-term dependencies and has long been a dominant preference in NLP. Recently, the pioneering work of ViT [2] has demonstrated that directly applying a pure transformer to a sequence of image patches brings exciting results on various image classification benchmarks with large-scale pre-training. The powerful modeling capability from ViT spawns many novel vision transformer models in computer vision. DeiT [23] makes use of a bunch of training techniques and particularly proposes a distillation procedure to release the ViT from heavy dependency on large datasets, outperforming the original ViT as well. Later, PVT [24] and Swin [3] introduce the pyramid structure into transformers to generate multi-scale feature maps, making it a unified backbone for multiple downstream vision tasks. Subsequently, many vision transformers [5, 25, 3, 6, 24, 26] are presented to improve the performance and achieve astonishing results on multiple benchmarks. Until now, ViTs have been widely applied across various vision tasks, including image classification [3, 7], object detection [3, 4], segmentation [27], point cloud [28], and so on. However, the computational cost of ViTs is still intensive and scales up quickly as numbers of MSA heads, embedding width [29].

### 2.2  CNN Pruning

Extensive efforts have been conducted to prune CNNs for better efficiency via weight magnitude [30], filter norm [31], scaling factor [32], learned performance proxy [12], etc. More relevantly, some works utilize Taylor expansion to approximate the loss of pruning. OBD [19] and OBS [20] aim to prune weights with the least error approximated by second-order Taylor derivatives, which is intractable for millions of parameters in modern deep models. [33] introduces an importance metric based on first-order Taylor expansion, avoiding the expensive computational cost of the Hessian matrix. L-OBS [9] proposes a layer-wise pruning method using a criterion based on second-order derivatives of a layer-wise error function. CCP [34] analyzes the layer-wise overall impact of pruned channels based on the second-order Taylor expansion when pruning each layer. [10] measures the importance by a squared difference of prediction errors and approximates it using Taylor expansion. In experiments, they show that the Hessian matrix makes negligible improvements, which indicates that the interactions are limited in pruning CNN.

### 2.3  Transformer Pruning

Compared to pruning CNNs, pruning transformers is still in the early stage. Some previous works put efforts into pruning different substructures, including attention head [13], basic block [35]. The recent advances in ViTs motivate many researchers to design specific methods for ViTs. Instead of training full ViTs, [8] integrates sparse training into the transformer to train the smaller subnetwork from scratch. Another line of works [14, 36, 37, 38] extract data-related redundancy and remove unnecessary image patches. Unfortunately, pruning patches breaks down the spatial structure of ViTs. Recently, NVP [16] first shows that pruning all components reaches better acceleration. UP [17] considers KL-Divergence change for each parameter on a proxy dataset, using the same compression ratio in all blocks. ViT-Slim [39] designs differentiable soft masks on each component and imposes $\ell_1$ sparsity constraint to force the mask to be sparse. Different from the hand-crafted pruning ratios in prior works, we first quantitatively analyze the interactions between components and adaptively learn the proper pruning ratio for each component by solving the optimization problem, offering the best flexibility to find the optimal subnetwork in a more collaborative manner.

## 3  Method

In this section, based on the relationship between computational cost and prunable ViT components, we present the scheme of pruning all components in ViTs comprehensively.

### 3.1  Components Analysis

To reveal how each component affects the computational cost of ViTs and analyze the trade-off between accuracy and FLOPs reduction for the network, we conduct experiments on pruning the

Table 1: Computation analysis. The last two columns give examples of FLOPs in the practical DeiT and Swin model. $n$ is the number of patches, $d_h$ is the feature dimension of each head.

| Part | Computation | DeiT-B | Swin-B |
|------|-------------|--------|--------|
| MSA Projects | $3ndhd_h + nhd_h d$ | 5.57G | 4.91G |
| MSA attention | $2n^2 hd_h$ | 0.72G | 0.31G |
| FFN | $2ndd_{ff}$ | 11.2G | 9.87G |
| Total | $2nhd_h(2d + n) + 2ndd_{ff}$ | 17.6G | 15.4G |

commonly used DeiT. Figure 2a clearly shows that pruning a single component results in the unsatisfactory performance of pruned models. Meanwhile, each component makes distinct contributions to the network. Furthermore, we break down the computation of ViT into three parts, while ViT variants follow a similar configuration. As shown in Table 1, MSA and Feed-Forward Network (FFN) account for 36% and 64% computation respectively. Combining with the architecture characteristics illustrated by Figure 1, structural pruning can be applied to three components to accelerate ViTs: 1) attention head, which only affects MSA computation; 2) hidden dimension, which only affects FFN; 3) embedding dimension, which is shared across both MSA and FFN. Despite embedding dimension being responsible for the largest amount of computation, it is more tricky to evaluate the importance of embedding dimension than other components. The reason behind this lies in that embedding dimensions of different linear layers are entangled due to skip connection. Consequently, collaboratively pruning all components needs to handle the difficulty in evaluating the importance of distinct components.

## 3.2 Collaborative Pruning

Based on the above observations, our collaborative pruning scheme adjusts all components together to achieve a better trade-off between accuracy and latency. Given a ViT model, we apply structural pruning on all components by removing unnecessary parameter groups that cause a minimum performance drop on the network. The performance drop can be reflected by the perturbation in the loss function $\mathcal{L}$. Inspired by OBD [19], we use the Taylor expansions to construct an approximated expression of the loss function and analytically estimate the joint influence of pruning. Formally, we define collaborative pruning as a kind of perturbation $\Delta \boldsymbol{w}$ on the whole weight vector $\boldsymbol{w}$, under a certain computational cost constraint $C_{budget}$ (e.g. FLOPs). The optimal subnetwork can be found through the following optimization problem:

$$\min_{\Delta \boldsymbol{w}} \Delta \mathcal{L} = \mathcal{L}(\boldsymbol{w} + \Delta \boldsymbol{w}) - \mathcal{L}(\boldsymbol{w}),$$
$$s.t. \ \ C(\boldsymbol{w} + \Delta \boldsymbol{w}) \leq C_{budget}, \tag{1}$$

where $C(\boldsymbol{w} + \Delta \boldsymbol{w})$ represents the computational cost of the pruned model and $\Delta \boldsymbol{w} = \boldsymbol{b} \odot \boldsymbol{w} - \boldsymbol{w}$. The binary mask $\boldsymbol{b}$ indicates whether each weight should be pruned and $\odot$ is Hadamard product, $\boldsymbol{b}_i = 0$ means the $i$-th weight should be pruned. $\Delta \mathcal{L}$ denotes the difference of loss function before and after pruning, which can be further formulated as follows:

$$\Delta \mathcal{L} = \Delta \boldsymbol{w}^T \boldsymbol{g} + \frac{1}{2} \Delta \boldsymbol{w}^T \boldsymbol{H} \Delta \boldsymbol{w} + O(||\Delta \boldsymbol{w}||^3), \tag{2}$$

where $\boldsymbol{g}$ and $\boldsymbol{H}$ are the gradient vector and the Hessian matrix w.r.t $\boldsymbol{w}$.

According to *quadratic approximation* [19], the third term in Equation 2 can be neglected. Generally, the first term $\Delta \boldsymbol{w}^T \boldsymbol{g}$ measures the impact of each parameter group playing individually. As we employ structural pruning, the first term can be further rewritten as the sum over perturbation of all pruned parameter groups:

$$\Delta \boldsymbol{w}^T \boldsymbol{g} = \sum_{s \in S} I_i. \tag{3}$$

where $I_i = \sum_{i \in s} w_i g_i$ is the individual importance for the pruned parameter group. $S$ is the set of all prunable parameter groups. $w_i$ and $g_i$ is the weight and gradient for the $i$-th parameter respectively in the parameter group $s$. And the $\Delta \boldsymbol{w}^T \boldsymbol{g}$ can be easily computed since $g_i$ is already available from backward propagation.

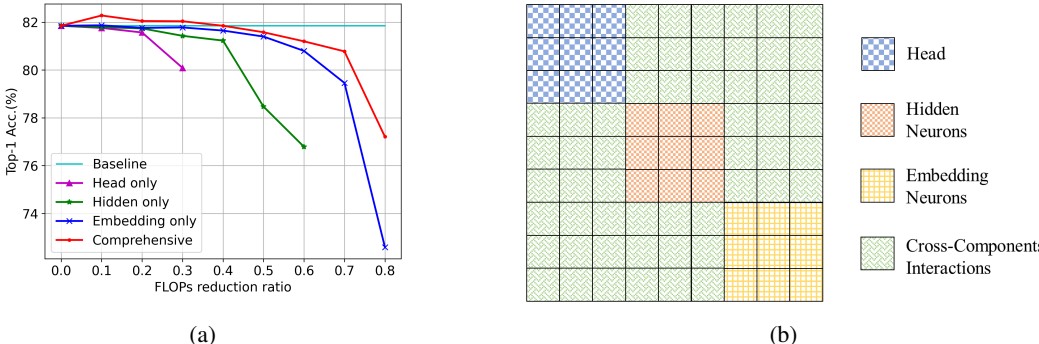

(a)                                                                    (b)

Figure 2: (a) Accuracy of models applied with different pruning strategies on ImageNet. The baseline is DeiT-Base (FLOPs reduction ratio is 0). $X$-only means the model is pruned along a single $X$ component. "Comprehensive" means employing pruning across all components. (b) An example illustration of the full Hessian matrix. Green blocks indicate the interactions between components. Other blocks mean the interactions within the corresponding component.

The second term in Equation 2 provides rich knowledge about the interactions between components, which is essential for pruning ViTs. However, the brute-force method suffers from huge memory space and computational cost required by the large-scale full Hessian matrix in Equation 2. This is infeasible on many devices and restricts the application of our method.

To tackle this problem, we propose a more efficient approximation. Due to the different roles that different components play in ViT, we split all interactions encoded by the full Hessian matrix into intra-component interactions (blue, orange, yellow parts in Figure 2b) and inter-component interactions (green parts in Figure 2b). Correspondingly, $\Delta \boldsymbol{w}$ during pruning can be partitioned into weight perturbations within component of head, hidden neurons and embedding neurons $[\Delta \boldsymbol{w^{(1)}}, \Delta \boldsymbol{w^{(2)}}, \Delta \boldsymbol{w^{(3)}}]$. Substituting the partitioned weight perturbations into the second term in Equation 2, we have

$$\frac{1}{2}\Delta \boldsymbol{w}^T \boldsymbol{H} \Delta \boldsymbol{w} = \frac{1}{2} \sum_{k,l=1}^{3} \Delta \boldsymbol{w}^{(k)^T} \boldsymbol{H}^{(kl)} \Delta \boldsymbol{w}^{(l)}, \tag{4}$$

where $\boldsymbol{H}^{(kl)}$ is the Hessian block matrix that describes the interactions between component $k$ and component $l$. For example, $\boldsymbol{H}^{(11)}$, illustrated by the blue block in Figure 2b, represents the interactions within heads, while the green block $\boldsymbol{H}^{(12)}$ indicates the interactions between heads and hidden neurons.

Denoting all the learnable weights belongs to component $k$ by $\boldsymbol{w}^{(k)}$, we re-formulate each Hessian block in Equation 4 as:

$$\Delta \boldsymbol{w}^{(k)^T} \boldsymbol{H}^{(kl)} \Delta \boldsymbol{w}^{(l)} = \sum_{i=1}^{N^{(k)}} \sum_{j=1}^{N^{(l)}} m_{ij}^{(kl)} w_i^{(k)} h_{ij}^{(kl)} w_j^{(l)}$$

$$= N^{(k)} N^{(l)} \overline{\boldsymbol{M}^{(kl)} \boldsymbol{w}^{(k)} \boldsymbol{H}^{(kl)} \boldsymbol{w}^{(l)}}, \tag{5}$$

where $N^{(k)}$ and $N^{(l)}$ are the total number of parameters in component $k$ and $l$ respectively. $m_{ij}^{(kl)}$ is the mask of the interaction between the $i$-th parameter in component $k$ and the $j$-th parameter in component $l$. $m_{ij}^{(kl)} = 1$ means the interaction is added into weight perturbation due to pruning of the corresponding parameter and $m_{ij}^{(kl)} = 0$ means the interaction should be neglected. $\overline{\boldsymbol{M}^{(kl)} \boldsymbol{w}^{(k)} \boldsymbol{H}^{(kl)} \boldsymbol{w}^{(l)}}$ denotes the mean of Hessian block correspond to the pruned network.

Without loss of generality, every pruning process can be regarded as an independent weight sampling from the network. Thus the Hessian block can be transformed into the following expression:

$$\Delta \boldsymbol{w}^{(k)^T} \boldsymbol{H}^{(kl)} \Delta \boldsymbol{w}^{(l)} \approx N^{(k)} N^{(l)} \overline{\boldsymbol{M}^{(kl)}} \ \overline{\boldsymbol{w}^{(k)} \boldsymbol{H}^{(kl)} \boldsymbol{w}^{(l)}}, \tag{6}$$

where $\overline{M^{(kl)}}$ denotes the mean of the mask matrix, $u^{(kl)} = \overline{w^{(k)}H^{(kl)}w^{(l)}}$ is the mean of the full Hessian block matrix term. In this paper, $u$ is computed by Monte Carlo method [40] to further reduce computational cost.

During pruning, $\overline{M^{(kl)}}$ is generated by:

$$\overline{M^{(kl)}} = \overline{(1 - b^{(k)})[1 - b^{(l)}]^T} = \overline{(1 - b^{(k)})} \,\overline{[1 - b^{(l)}]^T}, \tag{7}$$

$$\tag{8}$$

where $b^{(k)}$ denotes the binary mask vector for the $k$-th component and $\overline{(1 - b^{(k)})}$ is identical to the pruning ratio. We define the pruning ratios of attention head, hidden, embedding by $\rho^{(1)}$, $\rho^{(2)}$ and $\rho^{(3)}$ respectively:

$$\rho^{(k)} = \frac{||1 - b^{(k)}||_0}{N^{(k)}}, \ k = 1, 2, 3, \tag{9}$$

where $|| \cdot ||_0$ is the $\ell_0$-norm, $N^{(k)}$ is the number of all parameters in component $k$. So $\overline{M^{(kl)}}$ is equal to $\rho^{(k)}\rho^{(l)}$, and we can simplify the interactions as:

$$N^{(k)}N^{(l)}\overline{M^{(kl)}} \ \overline{w^{(k)}H^{(kl)}w^{(l)}} = N^{(k)}N^{(l)}\rho^{(k)}\rho^{(l)}u^{(kl)}. \tag{10}$$

Therefore, the loss perturbation in Equation 2 can be approximated as:

$$\Delta\mathcal{L} \approx \Delta w^T g + \frac{1}{2}\sum_{k,l=1}^{3} N^{(k)}N^{(l)}u^{(kl)}\rho^{(k)}\rho^{(l)}. \tag{11}$$

As described above, the $\Delta w$ can be partitioned into weight perturbations along each component: $\Delta w = [\Delta w^{(1)}, \Delta w^{(2)}, \Delta w^{(3)}]$. Summarizing all above, the final objective of collaborative pruning that minimizes the joint importance can be rewritten as follows:

$$\min_{\rho^{(1)},\rho^{(2)},\rho^{(3)}} \Delta w^T g + \frac{1}{2}\sum_{k,l=1}^{3} N^{(k)}N^{(l)}u^{(kl)}\rho^{(k)}\rho^{(l)},$$
$$s.t. \ C(w + \Delta w) \leq C_{budget},$$
$$\Delta w^{(k)} = Prune(\rho^{(k)}), \quad k = 1, 2, 3. \tag{12}$$

Here $Prune$ represents an operation to prune weights within each component individually. Specifically, for the $k$-th component with pruning ratio $\rho^{(k)}$, the $Prune$ will update the mask of the least important $N^{(k)}\rho^{(k)}$ weights as zero. Correspondingly, weight perturbation $\Delta w^{(k)}$ becomes $b^{(k)} \odot w^{(k)} - w^{(k)}$. To evaluate the importance of each weight, we apply the Fisher information metric [11]:

$$I_i = \frac{1}{N}\sum_{n=1}^{N} w_i^2 \frac{\partial^2 \mathcal{L}_n}{\partial w_i^2} = \frac{1}{N}\sum_{n=1}^{N}\left(w_i\frac{\partial\mathcal{L}_n}{\partial w_i}\right)^2. \tag{13}$$

where $N$ is the number of samples. Finally, the whole optimization problem can be solved by Evolutionary Algorithm (EA) [21]. We provide a complete view of the collaborative pruning in Algorithm 1. When dealing with the more complicated detection network, we regard the neck and detection head as new components. Then collaborative pruning leverages the interactions involving all components to trim down the network.

**Relations to CNN pruning methods using the Hessian matrix** As described in Section 2.2, L-OBS [9] and CCP [34] leverage the second-order derivatives for pruning homogeneous CNNs. However, they both fail to deal with networks containing multiple heterogeneous components such as ViTs since the structural characteristic of each component is quite different and the importance of different components is incomparable. Simply applying them on ViTs will cause non-optimal pruning ratios for different components. Furthermore, these layer-wise pruning methods discard important relationships between layers, for example, the relationship between MSA and FFN. The layer-wise design also needs to set pre-defined ratios for each layer, which is not optimal. Going beyond both, we establish the benefits of Hessian as a tool to capture the essential interactions across different components and different layers in pruning ViTs. Through solving the optimization problem, our work adaptively learns optimal pruning ratios for each component.

---
**Algorithm 1** Collaborative Pruning with EA
---
**Input**: Pre-trained model $T_o$, FLOPs constraint $C_{budget}$, dataset D, search iterations $E$, population size $Q$, components number $M$, fitness value $f$
**Output**: Optimal pruned model
---
 1: Initialize $Q$ samples $q_i = (\rho_i^{(1)}, \rho_i^{(2)}, ..., \rho_i^{(M)})$, $i = 1, 2, ..., Q$
 2: Calculate importance $I$ for all structural groups according to Equation 13 and $u^{(ij)}$, $i, j = 1, 2, ..., M$ according to Equation 6 on D
 3: **for** iteration $e = 1$ to $E$ **do**
 4:     **for** sample $i = 1$ to $Q$ **do**
 5:         **if** $C(q_i) < C_{budget}$ **then**
 6:             $f_i = 0$, continue
 7:         **end if**
 8:         **for** pruning ratio $\rho_i^{(j)}$ in current sample $q_i$ **do**
 9:             $Prune$ the corresponding component $j$ according to importance $I$
10:         **end for**
11:         Compute the $f$ according to Equation 11
12:     **end for**
13:     Keep the Top-k fittest samples
14:     Generate new samples by $Crossover$ and $Mutation$
15: **end for**
16: **return** Optimal pruned model
---

# 4 Experiment

## 4.1 Pruning DeiT

In this section, we first analyze the performance of the proposed pruning method on the DeiT family of different model sizes, i.e., DeiT-Base/Small/Tiny. Then we show the benefits of pruning equipped with knowledge distillation.

**Implementation Details** The pruning process is performed on the pre-trained DeiT [1] released from official implementation on ImageNet-1k [41]. As our method is a one-shot method, the whole pruning process is performed fast on a single GPU. After pruning, we fine-tune the pruned network using the same setting as DeiT [23] without warm-up.

**Results** We present summarized results on Table 2. On DeiT-Base, SAViT can steadily outperform previous state-of-the-art works under various complexity settings. Impressively, when reducing the model size by 50% FLOPs, SAViT achieves 0.32% improvement with 17% fewer FLOPs than S²ViTE-B and even surpasses the baseline by 0.7%. When we further increase the compression ratio to 70%, our approach still obtains competitive performance with only a 0.18% accuracy drop, outperforming the recently proposed UVC by 1.1%. The strong performance indicates that SAViT benefits a lot from the interactions. On the other hand, the pruned models by our method show a better trade-off between accuracy and efficiency than hand-crafted models as well as models based on Neural Architecture Search (NAS). For example, our compressed model obtains higher accuracy than T2T-ViT-24 with fewer FLOPs. A similar observation holds for DeiT-Small, where our method outperforms SSP and S²ViTE by a large margin. Furthermore, the proposed method also achieves the best top-1 accuracy when pruning smaller DeiT-Tiny. The consistently remarkable performance of pruning DeiT family demonstrates the proposed collaborative pruning algorithm works very well with different model complexity, suggesting strong potential for widespread applications.

### 4.1.1 Knowledge Distillation

For better performance, we also investigate fine-tuning with knowledge distillation. Using Deit-Base-Distilled [23] as baseline, we prune it into different model size, i.e., **SAViT-B/S/T**, as shown in Table 3. Then we fine-tune the pruned models with a simple knowledge distillation strategy. The fine-tuning configuration is detailed in Appendix B. Table 3 lists the results of our method and previous

---
[1]https://github.com/facebookresearch/deit

Table 2: Results of pruning DeiT Base/Small/Tiny on ImageNet-1k dataset. We compare the parameters, FLOPs, and Top-1 accuracy of the pruned model with various models. X-600e means training the baseline model for a longer schedule, i.e., 600 epochs following the original recipe in [23]. * means hand-crafted models with comparable FLOPs. † means NAS-based models. Others are pruned models.

| Model | Param.(M) | ($\downarrow$%) | FLOPs(G) | ($\downarrow$%) | Top-1 Acc. (%) | $\Delta$ |
|---|---|---|---|---|---|---|
| DeiT-B | 86.6 | - | 17.6 | - | 81.84 | - |
| DeiT-B-600e | 86.6 | - | 17.6 | - | 82.01 | +0.17 |
| T2T-ViT-24* [42] | 64.1 | 26.0 | 13.8 | 21.6 | 82.30 | +0.46 |
| PVT-L* [24] | 61.4 | 29.1 | 9.8 | 44.3 | 81.70 | -0.14 |
| AutoFormer-B† [43] | 54.0 | 37.6 | 11.0 | 37.5 | 82.40 | +0.56 |
| SSP-B [8] | 56.8 | 34.4 | 11.8 | 33.1 | 80.80 | -1.04 |
| $S^2$ViTE-B [8] | 56.8 | 34.4 | 11.8 | 33.1 | 82.22 | +0.38 |
| Evo-ViT [36] | 86.6 | 0.0 | 11.7 | 33.5 | 81.30 | -0.54 |
| EViT-DeiT-B [38] | 86.6 | 0.0 | 11.6 | 34.1 | 81.30 | -0.54 |
| DynamicViT [14] | 86.6 | 0.0 | 11.5 | 34.7 | 81.30 | -0.54 |
| ViT-Slim [39] | 52.6 | 39.3 | 10.6 | 39.8 | 82.40 | +0.56 |
| SAViT (ours) | 51.9 | 40.1 | 10.6 | 39.8 | **82.75** | **+0.91** |
| VTP-B [44] | 47.3 | 45.4 | 10.0 | 43.2 | 80.70 | -1.14 |
| PS-ViT-B [37] | 86.6 | 0.0 | 9.8 | 44.3 | 81.50 | -0.34 |
| SAViT (ours) | 42.6 | 50.8 | 8.8 | 50.0 | **82.54** | **+0.70** |
| UVC [45] | - | - | 8.0 | 54.5 | 80.57 | -1.27 |
| SAViT (ours) | 25.4 | 70.7 | 5.3 | 69.9 | **81.66** | **-0.18** |
| DeiT-S | 22.1 | - | 4.6 | - | 79.85 | - |
| DeiT-S-600e | 22.1 | - | 4.6 | - | 80.02 | +0.17 |
| AdaViT-S [15] | 22.1 | 0.0 | 3.6 | 21.7 | 78.60 | -1.25 |
| DynamicViT [14] | 22.1 | 0.0 | 3.4 | 26.1 | 79.60 | -0.25 |
| EViT-DeiT-S [38] | 22.1 | 0.0 | 3.0 | 34.8 | 79.50 | -0.35 |
| SSP-S [8] | 14.6 | 33.9 | 3.1 | 31.6 | 77.74 | -2.11 |
| $S^2$ViTE-S [8] | 14.6 | 33.9 | 3.1 | 31.6 | 79.22 | -0.63 |
| SAViT (ours) | 14.7 | 33.5 | 3.1 | 31.7 | **80.11** | **+0.26** |
| DeiT-T | 5.7 | - | 1.3 | - | 72.20 | - |
| SSP-T [8] | 4.2 | 26.3 | 0.9 | 23.7 | 68.59 | -3.61 |
| $S^2$ViTE-T [8] | 4.2 | 26.3 | 0.9 | 23.7 | 70.12 | -2.08 |
| SAViT (ours) | 4.2 | 25.2 | 0.9 | 24.4 | **70.72** | **-1.48** |

compression methods. As expected, assembling knowledge distillation boosts the performance of SAViT. Notably, SAViT achieves 77.0% accuracy with only 1.3G FLOPs, substantially reducing DeiT-Base-Distilled computational cost by 92.6% FLOPs and surpassing the distillation-only DeiT-Tiny-Distilled as well as NVP-T. This means a heavily pruned model from a larger model achieves better accuracy and indicates the necessity of pruning, consistent with the previous observations [47]. Besides, SAViT substantially improves accuracy over other compression methods.

## 4.2 Pruning Swin Transformer

To verify the generalization of our method, we also conduct experiments on a more challenging ViT, i.e., hierarchical Swin Transformer. It adopts the local windows attention mechanism to reduce the computational cost for downstream tasks and becomes a general-purpose backbone [3, 26]. We compare our approach with the baseline pruning method $\ell_1$-norm, which is extended from CNN compression [31] and belongs to importance-based methods.

**Implementation Details**  Similar to DeiT, we perform pruning on the official pre-trained Swin[2] on ImageNet-1k. Finally, we fine-tune the pruned network for 300 epochs under the same strategies as Swin [3].

---

[2]https://github.com/microsoft/Swin-Transformer

Table 3: Results of combining pruning and knowledge distillation for compressing DeiT-Base-Distilled. DeiT-B/S/T-Distilled indicates the pre-trained baseline from DeiT family [23]. UP-DeiT and NVP adopt pruning before knowledge distillation. The unpruned DeiT-B-Distilled is used as the teacher for distilling SAViT-B/S/T.

| Model | Param. | FLOPs | Top-1 Acc. |
|---|---|---|---|
| DeiT-B-Distilled | 86.6M | 17.6G | 83.36 |
| NVP-B [16] | 34.0M | 6.8G | 83.29 |
| SAViT-B (ours) | 33.6M | 6.7G | **83.31** |
| DeiT-S-Distilled | 22.1M | 4.6G | 81.20 |
| Manifold [46] | 22.1M | 4.6G | 81.48 |
| UP-DeiT-S [17] | 22.1M | - | 81.56 |
| NVP-S [16] | 21.0M | 4.2G | 82.19 |
| SAViT-S (ours) | 20.6M | 4.2G | **82.38** |
| DeiT-T-Distilled | 5.6M | 1.3G | 74.50 |
| Manifold [46] | 5.6M | 1.3G | 75.06 |
| UP-DeiT-T [17] | 5.7M | - | 75.79 |
| NVP-T [16] | 6.9M | 1.3G | 76.21 |
| SAViT-T (ours) | 6.6M | 1.3G | **76.95** |

**Results** Table 4 compares the results of our approach with the baseline pruning methods. Pruning Swin is more challenging since the elaborately designed hierarchical architecture has better parameter efficiency and less redundancy. Our approach can reduce the computational costs by 50% with a slight influence, consistently surpassing the baseline pruning method $\ell_1$-norm. The impressive performance of pruning Swin Transformer again demonstrates the superiority of the proposed pruning algorithm.

Table 4: Results of pruning Swin-Base on ImageNet-1k dataset.

| Model | Method | Param. | FLOPs | Top-1 Acc. |
|---|---|---|---|---|
| | Baseline | 87.8M | 15.4G | 83.51 |
| Swin-B | $\ell_1$-norm[31] | 36.0M | 7.8G | 80.95 |
| | ours | 33.0M | 7.7G | **82.62** |

## 4.3 Pruning Detection Network

Since the detection network consists of more components such as neck, it becomes harder to prune such complicated architecture. Benefiting from the collaborative optimization of our method, the proposed approach can be easily transferred to accelerate the detection network by adding the neck and detection head as new components. We employ pruning on the popular object detection framework Faster R-CNN [48] with Swin-Tiny backbone on COCO 2017 dataset [49] and report mean Average Precision (mAP) for comparison. To the best of our knowledge, we are the first to apply pruning to detection networks with ViT as the backbone. Besides, we extended $\ell_1$-norm pruning as our baseline method for comparison.

As noticed in Table 5, when targeting a considerable compression ratio (i.e., 58% parameters and 70% FLOPs reduction), our approach only has a slight mAP drop, outperforming the $\ell_1$-norm by a large margin, e.g. 3.1% mAP. This indicates that the proposed method can capture the interactions between multiple components in a more complicated network and excavate the redundancy well.

Table 5: Pruning Faster R-CNN detection network on COCO.

| Model | Method | Param. | FLOPs. | mAP |
|---|---|---|---|---|
| | Baseline | 45.2M | 221.7G | 45.5 |
| Faster R-CNN (Swin-T) | $\ell_1$-norm[31] | 21.7M | 67.8G | 42.1 |
| | ours | 19.2M | 67.8G | **45.2** |

### 4.4 Ablation Study

**Latency Measurement**    We strictly measure the latency of compressed ViTs using CUDA benchmark mode, please see Appendix C for details. Table 6 shows that the proposed method can bring 0.7% accuracy gains when compressing the DeiT-Base to achieve $2.0\times$ FLOPs reduction and $1.55\times$ inference speedup. More significantly, a larger $2.05\times$ inference speedup of DeiT-Base can be obtained with merely 0.18% loss of accuracy. We further provide operator-level speedup analysis in Appendix C. These observations suggest that our method can prune ViTs for better latency in practical applications.

Table 6: Run time speedup of compressed DeiT on Nvidia V100.

| Model | Param. ($\times$) | FLOPs ($\times$) | Speedup ($\times$) | Top-1 Acc.(%) |
|---|---|---|---|---|
| | 86.6M (1.00) | 17.6G (1.00) | 1.00 | 81.84 |
| DeiT-Base | 42.6M (2.03) | 8.8G (2.00) | 1.55 | 82.54 |
| | 25.4M (3.41) | 5.3G (3.32) | 2.05 | 81.66 |

**The Impact of Interactions**    An appealing feature of our method is the consideration of interactions between components, which helps identify redundancy accurately. To demonstrate its effectiveness, we prune DeiT-Base and Faster R-CNN according to the optimization function with and without cross-components interactions under the same acceleration target, i.e., 70% FLOPs reduction. For DeiT-Base, we fine-tune the pruned models for 80 epochs following the identical setting in Section 4.1. For Faster R-CNN, the pruned model is fine-tuned using the same setting as Section 4.3. Table 7 clearly shows that optimizing individual importance and interactions together gives the best result. Moreover, we observe that interactions help adjust all components toward a more balanced architecture. These observations suggest that interactions indeed play an important role in identifying structural redundancy and can boost the performance of the pruned model.

Table 7: Ablation study of pruning with and without cross-components interactions. $\rho_1 \sim \rho_5$ are the pruning ratios for the head, hidden dimension, embedding, Feature Pyramid Network, and detection head respectively.

| Model | Interactions | Pruning Ratio | | | | | Top-1 Acc./mAP |
|---|---|---|---|---|---|---|---|
| | | $\rho_1$ | $\rho_2$ | $\rho_3$ | $\rho_4$ | $\rho_5$ | |
| DeiT | ✗ | 0.13 | 0.12 | 0.69 | - | - | 79.68 |
| | ✓ | 0.42 | 0.40 | 0.52 | - | - | **80.78** |
| Faster R-CNN | ✗ | 0.55 | 0.68 | 0.61 | 0.72 | 0.61 | 44.5 |
| | ✓ | 0.55 | 0.53 | 0.71 | 0.66 | 0.52 | **45.2** |

## 5    Conclusion

In this paper, we present a versatile ViT accelerating framework that collaboratively prunes all components. Based on the theoretical analysis, we construct a Taylor-based optimization function to take full advantage of the interactions between heterogeneous components. As the Hessian matrix requires a huge computation cost, we derive an approximation to transform the Hessian matrix into pruning ratios and achieve fast pruning. Then the optimization problem is solved toward the optimal trade-off between accuracy and computational cost. We also show the proposed framework can be applied to prune more complicated architecture, e.g., detection network. Extensive experiments demonstrate that the proposed framework significantly reduces computational costs without compromising performance on various models as well as tasks.

## Acknowledgments and Disclosure of Funding

This work was supported by the National Natural Science Foundation of China (No. U19B2043).

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
