# A  Pruned Architecture

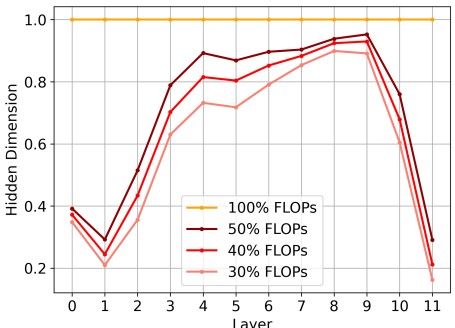

(a) Layer-wise hidden dimensions of the pruned model under different FLOPs targets.

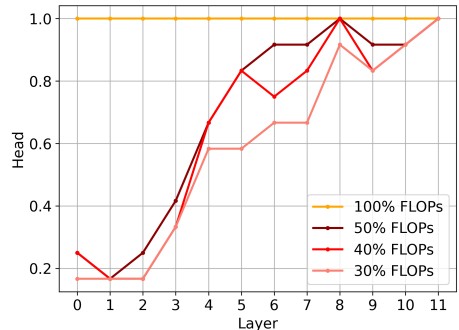

(b) Layer-wise head numbers of the pruned model under different FLOPs targets.

Here we analyze the parameter distribution of the pruned model from DeiT-Base. Figure 1a shows the reserved hidden dimension for each layer under different FLOPs targets, e.g. $50\% \sim 30\%$. All layers have the same hidden dimension of 3072 in the original DeiT. After pruning, we observe that the majority of the remaining parameters concentrate on the middle layers, while the lower layers require fewer parameters. We hypothesize that this occurs because the middle layers incorporate more global information than the lower layers and attempt to build more complex representations. Similarly, the results in Figure 1b show the reserved head number of MSA for each layer. We observe that more heads have remained in deeper layers. As work [1] mentions that in the first few layers, the heads tend to focus on local patterns while deeper layers attend to larger patterns. More heads in deeper layers help capture more complex patterns.

The above analysis can offer some insights for the community to design more efficient ViTs. As the uniform parameter distribution in the original DeiT contains a high degree of redundancy, it may not fully unleash the modeling capacity and representation flexibility of the model. We can construct more efficient ViT using the discovered guidelines. For example, we can keep a smaller hidden dimension ratio in lower layers instead of the same ratio in current ViTs.

# B  Knowledge Distillation

We consider the uncompressed model as a teacher for knowledge distillation. Let $\sigma$ be the softmax function, $z_s^{cls}$ and $z_s^{dist}$ the logits of the cls token and distillation token in student model respectively, $z_t^{cls}$ and $z_t^{dist}$ are the logits of teacher model, $\mathcal{L}_{KL}$ the Kullback–Leibler divergence loss, $\tau$ the temperature for the softmax. The teacher distillation loss is:

$$\mathcal{L}_{ds} = \tau^2 * (\mathcal{L}_{KL}(\sigma(z_s^{cls}/\tau), \sigma(z_t^{cls}/\tau)) + \mathcal{L}_{KL}(\sigma(z_s^{dist}/\tau), \sigma(z_t^{dist}/\tau))), \tag{1}$$

**Experiments**  We use the same training recipe as DeiT [2] unless otherwise mentioned. Specifically, the temperature $\tau$ is set to 0.05, the drop path rate is 0, and the learning rate warm-up is removed. Then we fine-tune the pruned model with an initial learning rate of $0.0003 \times \frac{batchsize}{512}$ for 300 epochs. We observe that applying only teacher distillation loss as a training objective achieves better performance than that of combining distillation loss and ground-truth loss, which is also observed in miniViT [3]. We guess this is due to the limited learning capacity of the tiny model. Compared to the ground-truth label, the more detailed knowledge from a teacher is easier for the tiny model to learn.

# C  Latency Measurement

**Experiment Setting**  Following [4], we measure the latency on an NVIDIA V100 GPU with a batch size of 256 images, using PyTorch on CUDA 10.1. To eliminate the impact of data loading, the time for data I/O is excluded. To reduce the impact of GPU warm-up, we conduct 100 forward passes. Then the median of the 1000 forward passes of the model during inference is reported. The setting is shared across original models and compressed models.

**Operator-level Speedup**   Furthermore, we break down the latency of baseline DeiT-Base and 70% FLOPs pruned model into operator levels. Table 1 shows the results. After pruning, the FLOPs remain 30% and the ideal speedup is 3.3x. As for the actual GPU latency speedup, it can be observed that the *matmul* achieves an almost ideal 3.04x speedup. However, LN, Softmax, and other memory-related operations can only reach 1.77x due to that these operations could not be reflected by FLOPs and are not linear w.r.t. FLOPs reduction. The above analysis shows that pruning can achieve ideal matmul operations speedup on GPUs (matmul computation is reflected by FLOPs). These are also observed in other ViT pruning works [5, 6].

Table 1: Operator-level run time speedup of compressed DeiT on Nvidia V100.

| Operators | Matmul. | LN | GELU | Softmax | Other memory-related ops | Total |
|---|---|---|---|---|---|---|
| DeiT Base latency(ms) | 125(38.9%) | 18.0(5.6%) | 17.8(5.5%) | 10.3(3.2%) | 151.5(46.9%) | 323(100%) |
| 70% FLOPs Pruned(ms) | 41.3(26.2%) | 11.7(7.4%) | 10.8(6.8%) | 6.0(3.8%) | 87.8(55.7%) | 101(100%) |
| Speedup | 3.04x | 1.54x | 1.65x | 1.72x | 1.73x | 2.05x |