# OpenReview forum: "SAViT: Structure-Aware Vision Transformer Pruning via Collaborative Optimization"
_NeurIPS.cc/2022/Conference — NeurIPS 2022 Accept_

### Official Review · Reviewer_HogS · 2022-07-09

**Rating:** 5
**Confidence:** 3
**Soundness:** 3 good
**Presentation:** 3 good
**Contribution:** 3 good

**Summary:**

This paper proposes SAViT,  a structure-aware pruning method for transformer-based architecture, which jointly prunes parameters in different components by considering interactions between these components. Experiments on different ViT architectures and vision tasks demonstrate the effectiveness of SAViT.

**Questions:**

See "Weaknesses"

**Limitations:**

The authors haven't discussed any limitations.

**Strengths And Weaknesses:**

Strengths
- The idea of collaboratively pruning all components of a model is interesting.
- The performance gain is impressive when compared to sufficient state-of-the-art methods.
- This paper is well-written and easy to follow.

Weaknesses
- It's not clear how much computation cost and data is needed for the proposed method during pruning, when compared to state-of-the-art network pruning methods.
- The main novelty of this work is the interaction of different components during pruning, so the ablation study on this design is important. In my understanding, Line 297-305 and Table 7 aim to give such ablation studies, while it's not clear whether the setting of "without second-order interations" (Line 300) in this ablation study means dropping all Hessian-based terms in Eq. (4) or only drop the cross-components terms (green blocks in Figure 2(b))? I think the latter one can better reflect the main contribution of the proposed method.

---

> ### Author Response · Authors · 2022-08-02
> **Response to Reviewer HogS**
>
> We are appreciated that you had a positive initial impression, and hope the responses below can solve your concerns.
> ## Q1
> Here we list the required computation cost and data of several state-of-the-art network pruning methods as well as ours. As for the data in our approach, we empirically find that pruning using 10\% training data works well as that using all training data.
>
> ***
> Method. &emsp;&emsp;&emsp;&emsp;&emsp;&emsp; Computation Cost(Search cost) &emsp;&emsp; Fine-tune cost &emsp;&emsp;&emsp; Data
> ***
> S$^{2}$ ViTE[1] &emsp;&emsp;&emsp;&emsp;&emsp;&emsp; &emsp;&emsp;&ensp;600 epochs &emsp;&emsp;&emsp;&emsp;&emsp;&emsp;&emsp;&emsp;&emsp; 0 epochs &emsp;&emsp; training dataset
>
> NVP[2] &emsp;&emsp;&emsp;&emsp;&emsp;&emsp;&emsp;&emsp;&emsp;&emsp; 10 epochs &emsp;&emsp;&emsp;&emsp;&emsp;&emsp;&emsp;&emsp;&emsp; 300 epochs &emsp;&ensp; training dataset
>
> ViT-Slim[3] &emsp;&emsp;&emsp;&emsp;&emsp;&emsp;&emsp;&emsp; 50 epochs &emsp;&emsp;&emsp;&emsp;&emsp;&emsp;&emsp;&emsp;&emsp; 300 epochs &emsp;&ensp; training dataset
>
> PS-ViT[4] &emsp;&emsp;&emsp;&emsp;&emsp;&emsp;&emsp;&emsp;&ensp; 300 epochs &emsp;&emsp;&emsp;&emsp;&emsp;&emsp;&emsp;&emsp;&ensp;&nbsp; 300 epochs &emsp;&ensp; training dataset
>
> Ours &emsp;&emsp;&emsp;&emsp;&emsp;&emsp;&emsp;&emsp;&emsp;&emsp;&emsp; <2 epochs &emsp;&emsp;&emsp;&emsp;&emsp;&emsp;&emsp;&emsp;&ensp; 300 epochs &emsp; 10\% training dataset
> ***
> Note that we train the 10\% training dataset for 12 epochs, other methods train on the whole training dataset, so the search time of our method is less than 2 epochs on the whole training dataset.
> ## Q2
> The ablation study in Table 7 means dropping the cross-components terms, as we aim to show the important impact of the interactions. We will address the ambiguity in the final version. Actually, we had conducted experiments dropping all Hessian-based terms and observed that the cross-component terms play a crucial role. The whole ablation experiments are summed up as follows.
>
> ***
> Model &emsp;&emsp;&emsp;&emsp;&emsp; Interactions &emsp;&emsp;&emsp;&emsp;&emsp;&emsp;&emsp;&emsp; Top-1 Acc.
> ***
> DeiT &emsp;&emsp; dropping all Hessian-based terms &emsp;&emsp;&emsp;79.56
>
> &emsp;&emsp;&emsp;&emsp;&ensp; dropping cross-component terms &emsp;&emsp;&emsp;79.68
>
> &emsp;&emsp;&emsp;&emsp;&emsp;keeping all Hessian-based terms  &emsp;&emsp;&emsp; **80.78**
> ***
>
> [1] Chen, T., et al. Chasing sparsity in vision transformers: An end-to-end exploration. In NeurIPS 2021.
>
> [2] Yang, H., et al. Nvit: Vision transformer compression and parameter redistribution. arXiv preprint 2021.
>
> [3] Arnav Chavan, et al. Vision transformer slimming: Multi-dimension searching in continuous optimization space. arXiv preprint 2022.
>
> [4] Yehui Tang, et al. Patch slimming for efficient vision transformers. arXiv preprint 2021.

---

### Official Review · Reviewer_2B6b · 2022-07-10

**Rating:** 6
**Confidence:** 3
**Soundness:** 3 good
**Presentation:** 3 good
**Contribution:** 4 excellent

**Summary:**

This paper presents a new neural network pruning method for vision transformers.  The proposed technique can effectively accelerate most vision transformers such as ViT and Swin Transformer by collaboratively pruning components such as multi-head self-attention, hidden neurons, and embedding neurons. Extensive experiments show that the proposed technique is efficient yet competitive in accuracy as compared with the state-of-the-art.

**Questions:**

No questions.

**Limitations:**

Yes

**Strengths And Weaknesses:**

The manuscript has the following pros as follows: 1) The paper has a clear motivation and the proposed technique is based on the theoretical analysis. 2) The paper is clearly written, and easy to follow in general; 3) A large number of experiments have been performed to validate the performance in various aspects.

I have some small concerns about this paper as listed. First, in Sec 4.4, the author said that the proposed method can bring accuracy gains when compressing the DeiT-Base. I think this conclusion is a bit inappropriate. As described in Sec. 4.1 and 4.2, the pruned models will further fine-tune 300 epochs, which means that the pruned models have a longer training schedule than baseline models. The accuracy gains may come from the longer training schedule.

In addition, the pseudo-code for the whole pruning algorithm in the Appendix is the core contribution of this work, which should be moved to the main paper.

---

> ### Author Response · Authors · 2022-08-02
> **Response to  Reviewer 2B6b**
>
> We are very glad you had a positive initial impression, and we provide pointwise responses to your concerns below.
> ## Q1
> To fairly compare against the state-of-the-art pruning methods, we fine-tuning 300 epochs after pruning following the existing studies on ViT compression [1,2]. Furthermore, to resolve your concerns, we train DeiT-Base/Small for 300+300 epochs as longer baselines following the original DeiT training recipe in the paper [3]. Actually, the performance of DeiT training saturates after 300\textasciitilde400 epochs. We report the results in the table below. Compares to longer baselines with 600 training epochs, our pruned models can still achieve 0.53\% accuracy gains on DeiT-Base and 0.09\% on DeiT-Small. These results suggest our pruning algorithm indeed brings accuracy gains.
> ***
> Models  &emsp;&emsp; 300 epochs  &emsp; 600 epochs &emsp; &emsp; Our pruned model
> ***
> DeiT-Base &emsp;&emsp;  81.84 &emsp;&emsp; &emsp; 82.01 &emsp;&emsp;&emsp; 82.54 (50.0\% FLOPs reduction)
>
> DeiT-Small &emsp;&ensp;  79.85  &emsp;&emsp;&emsp;&ensp; 80.02 &emsp;&emsp;&emsp; 80.11 (31.7\% FLOPs reduction)
> ***
> ## Q2
> Thanks for your advice, we will revisit and adjust the pseudo-code in a proper position in the final version.
>
> [1] Arnav Chavan, et al. Vision transformer slimming: Multi-dimension searching in continuous optimization space. arXiv preprint 2022.
>
> [2] Yehui Tang, et al. Patch slimming for efficient vision transformers. arXiv preprint 2021.
>
> [3] Touvron, H., et al. Training data-efficient image transformers \& distillation through attention. In ICML 2021.

---

> > ### Comment · Reviewer_2B6b · 2022-08-08
> > **Response to Authors**
> >
> > The authors' response addresses my concerns. I would like to keep my initial rating (weak accept) for this paper. Hope to see updates for the two mentioned issues in the final version.

---

### Official Review · Reviewer_ipJy · 2022-07-11

**Rating:** 5
**Confidence:** 4
**Soundness:** 2 fair
**Presentation:** 3 good
**Contribution:** 2 fair

**Summary:**

This paper presents a model pruning method for vision Transformers by jointly considering multiple possible pruning dimensions. To address the problem of joint optimization, a new collaborative pruning is designed. Experiments on multiple backbones (i.e., DeiT and Swin) and multiple tasks (i.e., ImageNet classification and COCO detection) show the effectiveness of the method.

**Questions:**

The paper presents a thorough study of the emerging area of efficient vision Transformers. The method is tested on multiple backbones and datasets. However, the joint optimization framework is similar to previous methods for CNNs, which makes the novelty of the pruning algorithm relatively low. Besides, I still have some concerns about the insufficient experiments, actual speedup on GPU, and reproducibility.  This paper can be stronger if the issues mentioned in the weaknesses subsection can be addressed.

**Limitations:**

Limitations of the proposed method are not discussed.

**Strengths And Weaknesses:**

Strengths:

- The idea of jointly considering multiple pruning dimensions is natural and well-motivated.

- The method is tested and works well on multiple tasks and backbones.

Weaknesses:

- The idea of joint optimization of the vision Transformer architecture is not very new. Many previous methods have explored the joint optimization problem for network acceleration. Recent work like AutoFormer also considers multiple dimensions for vision Transformers.

- Table 3 presents an important experiment to compare with previous state-of-the-art pruning methods. Since ViT/DeiT-B/S are considered as the standard models in many previous papers, it is better to provide the results on multiple model sizes (e.g., ViT-B/S/T) to clearly show the effectiveness of the proposed method.

- According to Table 6, pruning multiple dimensions may not lead to ideal actual speedup on GPUs.

- The method introduces a new pruning algorithm, which may not be easy to implement. Since the code is not available, I am a bit worried about the reproducibility of the method.


-----------------
Post rebuttal:

I would like to thank the authors for the detailed feedback and additional results. The response addressed my concerns about the insufficient experiments and actual speedup on GPUs. I would like to upgrade my rating to 5.

---

> ### Author Response · Authors · 2022-08-02
> **Response to Reviewer ipJy [Q3-Q4]**
>
> ## Q3
> The reason for the gap between the theoretical FLOPs compression rate and the actual speedup lies in that ViT contains many operators that affect running latency. Except for matrix multiplication/convolution operators, of which pruning aims to accelerate the computation, operations like LN and Softmax also require extra sophisticated computation and have a huge footprint on memory bandwidth. The extra computation and memory operation could not be reflected by FLOPs. To understand this, we break down the latency of baseline DeiT-Base and 70\% FLOPs pruned model into operator levels as follows.
>
> ***
> Operators &emsp;&emsp;&emsp;&emsp;&emsp;&emsp;&emsp; Matmul.  &emsp;&emsp;&emsp; LN  &emsp;&emsp;&emsp; GELU &emsp;&emsp;&emsp; Softmax &emsp; Other memory-related ops &emsp; Total
> ***
> DeiT Base latency(ms) &emsp; 125(38.9\%) &emsp;18.0(5.6\%) &emsp; 17.8(5.5\%) &emsp; 10.3(3.2\%) &emsp;&emsp;&emsp; 151.5(46.9\%) &emsp;&emsp;&emsp;&emsp;323(100\%)
>
> 70\% FLOPs Pruned(ms) &ensp; 41.3(26.2\%) &emsp;11.7(7.4\%) &emsp; 10.8(6.8\%) &emsp;&ensp;6.0(3.8\%) &emsp;&emsp;&emsp;&emsp;87.8(55.7\%) &emsp;&emsp;&emsp;&ensp;  157(100\%)
>
> Speedup &emsp;&emsp;&emsp;&emsp;&emsp;&emsp;&emsp;&emsp; 3.04x &emsp;&emsp;&emsp; 1.54x &emsp;&emsp;&emsp; 1.65x &emsp;&emsp;&emsp;&ensp; 1.72x &emsp;&emsp;&emsp;&emsp;&emsp;&emsp;&ensp; 1.73x  &emsp;&emsp;&emsp;&emsp;&emsp;&emsp; 2.05x
> ***
>
> After pruning, the FLOPs remain 30\% and the ideal speedup is 3.3x. As for the actual GPU latency speedup, it can be observed that **the matmul achieves an almost ideal 3.04x speedup**. However, LN, Softmax, and other memory-related operations can only reach 1.77x due to that these operations could not be reflected by FLOPs and are not linear w.r.t. FLOPs reduction. The above analysis shows that pruning can achieve ideal matmul speedup on GPUs (matmul computation is reflected by FLOPs). These are also observed in other ViT pruning works [10,11]. In fact, we are working on accelerating these operators like LN, Softmax, GELU, and other memory-related operators to achieve better speedup.
> ## Q4
> We promise that we will release the code in paper and checklist. Due to the double-blind reviewing rule of NeurIPS, we are not allowed to release the code now. We will release the code as soon. In addition, we have provided pseudo-code for implementation.
>
> [1] Molchanov, P., et al. Importance estimation for neural network pruning. In CVPR 2019.
>
> [2] Liu, L., et al. . Group fisher pruning for practical network compression. In ICML 2021.
>
> [3] Dong, X., et al. Learning to prune deep neural networks via layer-wise optimal brain surgeon. In NeurIPS 2017.
>
> [4] Chen, M. et al. Autoformer: Searching transformers for visual recognition. In ICCV 2021.
>
> [5] Yang, H., et al. Nvit: Vision transformer compression and parameter redistribution. arXiv preprint 2021.
>
> [6] Cai, H., et al. Once-for-All: Train One Network and Specialize it for Efficient Deployment. In ICLR 2019.
>
> [7] Ding Jia, et al. Efficient vision transformers via fine-grained manifold distillation. arXiv preprint 2021.
>
> [8] Hao Yu, et al. A unified pruning framework for vision transformers. arXiv preprint 2021.
>
> [9] Liu, Z., et al. Swin transformer: Hierarchical vision transformer using shifted windows. In CVPR 2021.
>
> [10] Chen, T., et al. Chasing sparsity in vision transformers: An end-to-end exploration. In NeurIPS 2021.
>
> [11] Yin, H., et al. A-ViT: Adaptive Tokens for Efficient Vision Transformer. In CVPR 2022.

---

> ### Author Response · Authors · 2022-08-02
> **Response to Reviewer ipJy [Q2]**
>
> ## Q2
> To show the effectiveness of our method, we conduct experiments on several model sizes and list the results below. DeiT-B-Distilled is adopted as the distillation teacher.
> ***
> Model &emsp;&emsp;&emsp;&emsp;&emsp;&emsp;&emsp;&emsp;&emsp;&emsp; Param.  &emsp;&emsp; FLOPs  &emsp;&emsp; Top-1 Acc.
>
> ***
>
> DeiT-B-Distilled(teacher) &emsp;&emsp;&ensp; 87M &emsp;&emsp;&ensp; 17.6G &emsp;&emsp;&emsp; 83.36
>
> NVP-B  &emsp;&emsp;&emsp;&emsp;&emsp;&emsp;&emsp;&emsp;&emsp;&emsp;&emsp; 34M &emsp;&emsp;&ensp; 6.8G &emsp;&emsp;&emsp; 83.29
>
> SAViT-B(ours) &emsp;&emsp;&emsp;&emsp;&emsp;&emsp;&emsp;&ensp; 33M &emsp;&emsp;&emsp; 6.7G &emsp;&emsp;&emsp; **83.31**
>
> ***
> DeiT-S-Distilled &emsp;&emsp;&emsp;&emsp;&emsp;&emsp;&nbsp; 22M &emsp;&emsp;&emsp; 4.6G &emsp;&emsp;&emsp; 81.20
>
> Manifold[7] &emsp;&emsp;&emsp;&emsp;&emsp;&emsp;&emsp;&emsp;  22M &emsp;&emsp;&emsp; 4.6G &emsp;&emsp;&emsp;&nbsp;81.48
>
> UP-DeiT[8] &emsp;&emsp;&emsp;&emsp;&emsp;&emsp;&emsp;&emsp;&nbsp; 22M &emsp;&emsp;&emsp;&emsp; - &emsp;&emsp;&emsp;&ensp;&nbsp; 81.56
>
> NVP-S[5] &emsp;&emsp;&emsp;&emsp;&emsp;&emsp;&emsp;&emsp;&emsp; 21M &emsp;&emsp;&emsp; 4.2G &emsp;&emsp;&emsp;&nbsp; 82.19
>
> SAViT-S(ours) &emsp;&emsp;&emsp;&emsp;&emsp;&emsp;&emsp; 21M &emsp;&emsp;&emsp; 4.2G &emsp;&emsp;&emsp; **82.38**
> ***
> DeiT-T-Distilled &emsp;&emsp;&emsp;&emsp;&emsp;&emsp; 5.6M &emsp;&emsp;&emsp; 1.3G &emsp;&emsp;&emsp;&ensp; 74.5
>
> Manifold[7] &emsp;&emsp;&emsp;&emsp;&emsp;&emsp;&emsp;&ensp;&nbsp; 5.6M &emsp;&emsp;&emsp; 1.3G &emsp;&emsp;&emsp;&ensp;&nbsp;75.1
>
> UP-DeiT[8] &emsp;&emsp;&emsp;&emsp;&emsp;&emsp;&emsp;&emsp; 5.7M &emsp;&emsp;&emsp;&emsp; - &emsp;&emsp;&emsp;&ensp;&emsp; 75.8
>
> NVP-T[5]  &emsp;&emsp;&emsp;&emsp;&emsp;&emsp;&emsp;&emsp;&emsp; 6.9M &emsp;&emsp;&emsp; 1.3G &emsp;&emsp;&emsp;&ensp; 76.2
>
> SAViT-T(ours) &emsp;&emsp;&emsp;&emsp;&emsp;&emsp;&emsp; 6.6M &emsp;&emsp;&emsp; 1.3G &emsp;&emsp;&emsp;&emsp;**77.0**
> ***
> As we can see, the more FLOPs we prune, the larger accuracy gap our method obtains over other state-of-the-art approaches. We notice that SAViT-B performs very close to the distillation teacher DeiT-B-Distilled and reaches almost the performance ceiling. In addition, we observe that a smaller model should use a smaller drop path rate, in accordance with Swin[9]. So we adjust the drop path rate for fine-tuning the pruned models as detailed in Appendix.

---

> ### Author Response · Authors · 2022-08-02
> **Response to Reviewer ipJy [Q1]**
>
> We appreciate your consideration and thoughtful feedback. We provide pointwise responses to your concerns below.
> ## Q1
> We dig into the importance of the interactions between components from a theoretical perspective. Based on the analysis, we derive to take advantage of the Hessian matrix to explicitly represent the interactions and prune the ViT automatically.
> Compared to previous methods for CNNs [1,2,3], our method is different in:
> **1)** The effect of interactions in CNNs and ViTs is different. As CNNs consist of homogeneous components, most works [1,2] drop the interactions and just apply individual importance, which has achieved pretty good performance. ViTs are significantly different from CNNs, and we show that cross-component interactions play a crucial role.
> **2)** The approximations of Hessian matrix are different. Due to the required huge memory of whole Hessian matrix, other works [3] compute the layer-wise Hessian matrix without considering cross-layer interactions. In contrast, we propose an efficient algorithm to approximate the global Hessian matrix. In a word, directly applying these approaches to prune ViTs is infeasible.
>
> On the other hand, the joint optimization of ViT can be categorized into Neural Architecture Search(NAS) and pruning. NAS considers interactions in a black-box-like, implicit form. Take AutoFormer [4] as an example, our method differs from it in: **1)** AutoFormer uses the classification accuracy of subnets from the supernet to implicitly reflect the interactions, while ours derives a theoretical term for the interaction explicitly. **2)** Since AutoFormer considers interactions in a implicit way, it consumes much more search time. Specifically, AutoFormer needs to train the supernet hundreds of epochs and evaluate thousands of subnets on the validation set. Ours starts from the pre-trained model and prunes it fast by the approximated loss perturbation. **3)** AutoFormer needs to design a discrete search space by hand. It searches QKV Dim in the range of (528, 624) by a step of 48. Instead, we can automatically search for a fine-grained and more suitable embedding dim.
>
> As for related ViT pruning works [5], they ignore the necessary interactions between components and require some hand-crafted parameters to balance the pruning ratios for every component.
>
> We list a thorough comparison of the related methods and ours below.
>
> ***
> Category &emsp;&emsp; Method &emsp; Application &emsp; Interactions &emsp;&emsp; Search Time &emsp;&ensp; Search Granularity &emsp;&emsp; Pattern
> ***
> NAS &emsp;&emsp;&emsp;&emsp; OFA[6] &emsp;&emsp;&emsp; CNN &emsp;&emsp;&emsp;&ensp; Implicit &emsp;&emsp;&emsp;&nbsp; 300 epochs &emsp;&emsp;&emsp;&emsp; Coarse &emsp;&emsp;&emsp;&emsp;  Hand-crafted
>
> &emsp;&emsp;&emsp;&emsp;&ensp; AutoFormer[4] &emsp;&ensp; ViT &emsp;&emsp;&emsp;&emsp; Implicit &emsp;&emsp;&emsp;&nbsp; 500 epochs &emsp;&emsp;&emsp;&emsp; Coarse &emsp;&emsp;&emsp;&emsp;  Hand-crafted
> ***
> Pruning &emsp;&ensp; Taylor-FO[1] &emsp;&ensp; CNN &emsp;&emsp;&emsp;&emsp;&ensp; No &emsp;&emsp;&emsp;&emsp;&ensp; 30 epochs &emsp;&emsp;&emsp;&ensp; Fine-grained &emsp;&emsp; Hand-crafted
>
> &emsp;&emsp;&emsp;&emsp;&emsp;&emsp; NVP[5] &emsp;&emsp;&emsp;&ensp; ViT &emsp;&emsp;&emsp;&emsp;&ensp;&ensp; No &emsp;&emsp;&emsp;&emsp;&ensp; 10 epochs &emsp;&emsp;&emsp; &ensp;Fine-grained &emsp; &emsp; Hand-crafted
>
> &emsp;&emsp;&emsp;&emsp;&emsp;&emsp; Ours &emsp;&emsp;&emsp;&emsp;&ensp; ViT &emsp;&emsp;&emsp;&emsp;&nbsp; Explicit  &emsp;&emsp;&emsp;&ensp; <2 epochs &emsp;&emsp;&emsp;&ensp;Fine-grained &emsp;&emsp; &ensp;Automatical
> ***
> Note that we conduct pruning with the 10\% training dataset for 12 epochs, while other methods do it on the whole training dataset, so the search time of our method is less than 2 epochs on the whole training dataset.

---

> ### Comment · Reviewer_ipJy · 2022-08-08
> **Thanks for the response**
>
> I would like to thank the authors for the detailed feedback and additional results. The response addressed my concerns about the insufficient experiments and actual speedup on GPUs. I am glad to see the search process of the method is much faster than the existing method. I raised the score to 5.

---

### Official Review · Reviewer_Mowb · 2022-07-13

**Rating:** 4
**Confidence:** 5
**Soundness:** 2 fair
**Presentation:** 2 fair
**Contribution:** 2 fair

**Summary:**

This work proposes to prune ViT from all components comprehensively, which considers the interactions between different components in pruning ViTs. Different from the homogeneous components of CNN, the components of ViT are always heterogeneous, thus this work constructs a Taylor-based optimization function to take full advantage of the interactions between heterogeneous components. To avoid the huge computation cost of the Hessian matrix, this work derives an approximation to transform the Hessian matrix into pruning ratios. Finally, it solves the optimization target towards the optimal trade-off between accuracy and computational cost. This work is validated on DeiT and Swin on ImageNet and also on detection experiments.

**Questions:**

Overall, I think this work has a reasonable motivation and good idea, but the definition of ∆w confused me.

**Limitations:**

Please refer to Weaknesses.

**Strengths And Weaknesses:**

Strengths:
1. This work is well-written and easy to follow.
2.  The motivation of this work is clear.


Weaknesses：

1. Please verify the definition of ∆w, as ∆w = b ⊙ w − w, while the optimization target (eq.1) is: min ∆L = L(w + ∆w) − L(w), thus ∆L = L(w + ∆w) − L(w) = L(b ⊙ w) - L(w). And  C(w + ∆w) = C(b ⊙ w). I think this target is not correct, which maybe cuased by the definition of ∆w.
2. Based on the definition of ∆w, I think we cannot get eq2 from eq1.
3. The pruning results on Swin are directly trained on ImageNet-1k, while Swin can achieve much better performance after pretraining on ImageNet-22k. So the author should also provide experimental results on it. No need to train Swin on ImageNet-22k, but directly load the pretrained weights on ImageNet-22k then do pruning along with finetuning on ImageNet-1k.

---

> ### Author Response · Authors · 2022-08-02
> **Response to Reviewer Mowb**
>
> Thanks for your sincere comments, we hope to have a further discussion to see if our response solves the concerns.
>
> ## Q1 \& Q2
> The target of eq.1 is correct and also used in related literature [1,2]. Here we provide a more detailed explanation. Assume the pre-trained model weight vector as $ w \in R^{N}$ , $N$ is the number of parameters, and the binary mask vector $b \in R^{N}$ is defined as follows:
>
> $b_i$=  1 &ensp; if the $i$-th weight is not pruned, 0 else the $i$-th weight is pruned.
>
> So we have the weight vector for the pruned model $b\odot w$. Correspondingly, the FLOPs computation for the pruned model is $C(b\odot w)$, and **the weight change before and after pruning is defined as $\Delta w = b\odot w - w$**. According to the Taylor series, the target (eq.1) has a series representation about $w$:
>
> $\Delta L = L(w+ \Delta w) - L(w)  = L(b\odot w)	- L(w) $
>
> &emsp;&emsp;$= L(w) + (b\odot w - w)^Tg + \frac{1}{2}(b\odot w - w)^TH(b\odot w - w) + O(||b\odot w - w||^3) - L(w)$
>
> &emsp;&emsp;$= (b\odot w - w)^Tg + \frac{1}{2}(b\odot w - w)^TH(b\odot w - w) + O(||b\odot w - w||^3)$
>
> &emsp;&emsp;$ = {\Delta w}^Tg+ \frac{1}{2}{\Delta w}^TH\Delta w + O(||\Delta w||^3)$
>
> ## Q3
> To fairly compare against other pruning methods, we conduct pruning on the Swin that directly trianed on ImageNet-1k. The pruning results have shown our algorithm can compress Swin with a slight performance drop. By setting with the ImageNet-22k pre-training recipe, Swin just trains with more data without any modifications to the network structure. Thus our method can naturally compress the ImageNet-22k pre-trained model. We conduct pruning on the Image-22k pre-trained model according to the configuration of Swin paper [3] and report the results below, which again demenstrates the effectiveness of our approach.
> ***
> Model&emsp;&emsp;&emsp;&emsp;&emsp;&emsp;&emsp;&emsp;&emsp;&emsp;&emsp;&emsp;&emsp;&emsp;Method&emsp;&emsp;FLOPs&emsp;&emsp;Top-1 Acc.
> ***
> ImageNet-1k directly Swin-B &emsp;&emsp;&emsp;Baseline[3] &emsp;15.4G &emsp;&emsp;&ensp; 83.5
>
> &emsp;&emsp;&emsp;&emsp;&emsp;&emsp;&emsp;&emsp;&emsp;&emsp;&emsp;&emsp;&emsp;&emsp;&emsp;&emsp;&emsp;SAViT &emsp;&emsp;&emsp; 7.7G &emsp;&emsp;&emsp; 82.6
>
> ***
> ImageNet-22k pre-trained Swin-B &ensp; Baseline[3] &nbsp; 15.4G &emsp;&emsp;&emsp; 85.2
>
> &emsp;&emsp;&emsp;&emsp;&emsp;&emsp;&emsp;&emsp;&emsp;&emsp;&emsp;&emsp;&emsp;&emsp;&emsp;&emsp;&emsp; SAViT &emsp;&emsp;&ensp;  7.7G &emsp;&emsp;&emsp;&ensp; 84.1
> ***
> [1] LeCun, Y., et al. Optimal brain damage. In NeurIPS 1989.
>
> [2] Peng, H., et al. Collaborative channel pruning for deep networks. In ICML 2019.
>
> [3] Liu, Z., et al. Swin transformer: Hierarchical vision transformer using shifted windows. In ICCV 2021.

---

### Author Response · Authors · 2022-08-08
**Dear Reviewers, Have We Addressed Your Concerns?**

Dear Reviewers,

We sincerely thank your time for the review, and we really hope to have a further discussion with you to see if our response solves your concerns before the end of discussion period. Thank you!

Best regards

---

### Meta-Review · Area_Chair_KUKZ · 2022-08-26

**Recommendation:** Accept
**Confidence:** Certain

**Metareview:**

The paper received three positive reviews and one negative review. The raised issues contain technical correctness, ImageNet-22K pertaining, insufficient experiments and speedup on GPUs, computational cost, clarity on ablation studies. During the rebuttal and discussion phases, most of the issues are addressed and reviewers are willing to upgrade. After checking all the reviews, rebuttals, and discussions, the AC agrees with the reviewers that the raised issues are well addressed. The authors shall revise according to the suggestions to further improve the current manuscript in the camera-ready submission. Also, the comparison to token selection-based ViT acceleration methods [a] shall be included in the experiments.

[a]. Not All Patches Are What You Need: Expediting Vision Transformers via Token Reorganizations. Liang et al. ICLR 2022.

**Award:**

No

---

### Decision · Program_Chairs · 2022-09-14

Accept